# Diurnal regulation of metabolism by G$_S$-alpha in hypothalamic QPLOT neurons

Kevin D. Gaitonde[1,2,3,4], Mutahar Andrabi[1,2,5], Courtney A. Burger[1,2,5], Shane P. D'Souza[1,2,3], Shruti Vemaraju[1,2,5], Bala S. C. Koritala[6,7], David F. Smith[6,7,8,9], Richard A. Lang[1,2,5]*

1 Division of Pediatric Ophthalmology, Abrahamson Pediatric Eye Institute, Visual Systems Group, Cincinnati Children's Hospital Medical Center, Cincinnati, OH, United States of America, 2 Science of Light Center, Cincinnati Children's Hospital Medical Center, Cincinnati, OH, United States of America, 3 Molecular & Developmental Biology Graduate Program, University of Cincinnati, College of Medicine, Cincinnati, OH, United States of America, 4 Medical Scientist Training Program, University of Cincinnati, College of Medicine, Cincinnati, OH, United States of America, 5 Department of Ophthalmology, College of Medicine, University of Cincinnati, Cincinnati, OH, United States of America, 6 Division of Pediatric Otolaryngology–Head and Neck Surgery, Cincinnati Children's Hospital Medical Center, Cincinnati, OH, United States of America, 7 Department of Otolaryngology–Head and Neck Surgery, University of Cincinnati College of Medicine, Cincinnati, OH, United States of America, 8 Division of Pulmonary Medicine, Cincinnati Children's Hospital Medical Center, Cincinnati, OH, United States of America, 9 The Center for Circadian Medicine, Cincinnati Children's Hospital Medical Center, Cincinnati, OH, United States of America

* Richard.lang@cchmc.org

**Data Availability Statement:** All relevant data are within the manuscript and its Supporting Information files.

## Abstract

Neurons in the hypothalamic preoptic area (POA) regulate multiple homeostatic processes, including thermoregulation and sleep, by sensing afferent input and modulating sympathetic nervous system output. The POA has an autonomous circadian clock and may also receive circadian signals indirectly from the suprachiasmatic nucleus. We have previously defined a subset of neurons in the POA termed QPLOT neurons that are identified by the expression of molecular markers (*Qrfp*, *Ptger3*, *LepR*, *Opn5*, *Tacr3*) that suggest receptivity to multiple stimuli. Because *Ptger3*, *Opn5*, and *Tacr3* encode G-protein coupled receptors (GPCRs), we hypothesized that elucidating the G-protein signaling in these neurons is essential to understanding the interplay of inputs in the regulation of metabolism. Here, we describe how the stimulatory G$_S$-alpha subunit (*Gnas*) in QPLOT neurons regulates metabolism in mice. We analyzed *Opn5$^{cre}$; Gnas$^{fl/fl}$* mice using indirect calorimetry at ambient temperatures of 22°C (a historical standard), 10°C (a cold challenge), and 28°C (thermoneutrality) to assess the ability of QPLOT neurons to regulate metabolism. We observed a marked decrease in nocturnal locomotion of *Opn5$^{cre}$; Gnas$^{fl/fl}$* mice at both 28°C and 22°C, but no overall differences in energy expenditure, respiratory exchange, or food and water consumption. To analyze daily rhythmic patterns of metabolism, we assessed circadian parameters including amplitude, phase, and MESOR. Loss-of-function GNAS in QPLOT neurons resulted in several subtle rhythmic changes in multiple metabolic parameters. We observed that *Opn5$^{cre}$; Gnas$^{fl/fl}$* mice show a higher rhythm-adjusted mean energy expenditure at 22°C and 10°C, and an exaggerated respiratory exchange shift with temperature. At 28°C, *Opn5$^{cre}$; Gnas$^{fl/fl}$* mice have a significant delay in the phase of energy expenditure and respiratory exchange. Rhythmic analysis also showed limited increases in rhythm-adjusted means of food and

**Funding:** This work was funded by National Eye Institute grants R01EY032029, R01EY032752, and R01EY032566 to RAL, and 5K08HL148551 and the American College of Surgeons / The Triological Society Clinical Scientist Development Award to DFS. The funders had no role in study design, data collection and analysis, decision to publish, or preparation of the manuscript.

**Competing interests:** The Lang lab participates in a sponsored research agreement between CCHMC and BIOS Lighting, Inc. This does not alter our adherence to PLOS ONE policies on sharing data and materials.

water intake at 22˚C and 28˚C. Together, these data advance our understanding of $G_{\alpha s}$-signaling in preoptic QPLOT neurons in regulating daily patterns of metabolism.

## Introduction

Many biological processes follow 24-hour cycles. These daily rhythms are influenced by the circadian clock oscillatory activity of the suprachiasmatic nucleus (SCN) [1]. In mice, the SCN regulates nocturnal patterns of locomotion and food consumption. Though the SCN contains an autonomous clock, the phase of this clock is entrained by environmental stimuli. A major entrainment stimulus is environmental light, as detected by the melanopsin-expressing intrinsically photosensitive retinal ganglion cells of the eyes [2]. Food [3] and sleep [1,4] can also influence SCN neuron activity. The ability of the SCN clock to entrain to several signals enhances the efficiency of metabolism and behavior to coordinate with daily variation in environmental signals.

In homeotherms, the preoptic area (POA) is also a central regulator of metabolic and behavioral processes, including thermoregulation and sleep [5]. POA neurons receive direct thermosensory input from the lateral parabrachial nucleus (LPB) [6] and project to other thermoregulatory areas including the dorsomedial hypothalamus and raphe pallidus [6]. Additionally, the POA receives indirect rhythmic input from the SCN through the DMH [7,8], and from extra-SCN retinal ganglion cells that project directly to the POA [9]. The POA also contains an autonomous clock that is thermosensitive [10], indicating a role for environmental sensitivity in the POA. In this manner, the POA integrates various rhythmic signals as it effects thermoregulatory changes.

The POA regulates energy expenditure (EE) to maintain homeostasis. Energy is expended through physical activity, the thermic effect of food intake, and cold-induced thermogenesis [11], all of which demonstrate robust circadian rhythms and sensitivity to temperature [12–17]. At cold ambient temperatures, the sympathetic nervous system upregulates thermogenesis in brown adipose tissue [18], cutaneous vasoconstriction reduces heat loss [19], and locomotion [20] and food intake [16] are increased to generate heat. At warm temperatures, the POA suppresses adaptive thermogenesis [21,22] and promotes heat dissipation by cutaneous vasodilation [21], while locomotion and food intake are decreased [16]. The metabolic adaptations required to respond to the daily variation in ambient temperature suggests that the POA may regulate circadian patterns of metabolism.

A population of excitatory POA neurons that demonstrate thermoregulation has been identified [23,24]. Through sequencing analysis, we have identified distinct molecular markers that define these POA neurons and may explain how they can respond to hormones or environmental stimuli [23]. These neurons have been termed QPLOT neurons (pyroglutamylated Arg-Phe-amide peptide *Qrfp*, prostaglandin EP3 receptor *Ptger3*, leptin receptor *LepR*, violet light sensing Opsin 5 *Opn5*, and the neurokinin B receptor NK3R encoded by *Tacr3*) to reflect the diversity of expressed molecular markers and signaling receptors. Manipulation of the activity of these neurons [6,25–27] produces immediate and marked changes in body temperature and EE, suggesting that QPLOT neurons are powerful regulators of metabolism.

Similarly, exposure of POA neurons expressing EP3, OPN5, or NK3R to their respective ligands produces changes in body temperature [6,28,29]. These proteins are G-protein coupled receptors that convert extracellular signals into intracellular guanine nucleotide-modulated second messenger cascades. Thus, understanding G-protein signaling in these neurons is

essential to elucidating the interplay of the various molecular inputs. The stimulatory $G_{\alpha s}$ subunit, which generates cyclic AMP through adenylyl cyclase, is especially interesting because it activates neurons and may explain violet light-mediated cAMP increases in *Opn5* neurons [6]. Furthermore, although the leptin receptor LEPR (*LepR*) is not a GPCR, leptin signaling in *Lepr* neurons in other hypothalamic areas may require or be inhibited by $G_{\alpha s}$-signaling [30,31]. Thus, analysis of $G_{\alpha s}$-signaling in QPLOT neurons is important to understanding the ability of QPLOT neurons to integrate incoming signals and modulate metabolism.

Here, we hypothesize that $G_{\alpha s}$-signaling activates QPLOT neurons to decrease metabolism. We have taken advantage of the selectivity of *Opn5^cre^* to conditionally delete the *Gnas* allele (encoding GNAS/$G_{\alpha s}$) in QPLOT neurons. In control and experimental cohorts, we have assessed the metabolic consequences of loss of *Gnas* in QPLOT neurons over multiple days and over a range of temperatures. We identify a role for $G_{\alpha s}$-signaling in QPLOT neurons that influences the central control of metabolism.

## Methods

### Animals

All procedures were conducted in accordance with protocols approved by the Institutional Animal Care and Use Committee at Cincinnati Children's Hospital Medical Center (protocol number 2021–0026). This study is compliant with all relevant ethical regulations regarding animal research. Mice were housed in a pathogen-free vivarium maintained at an ambient temperature of 22˚C and relative humidity of 30–70% with standard fluorescent lighting. Mice were placed on a normal chow diet (29% protein, 13% fat and 58% carbohydrate kcal; LAB Diet 5010) *ad libitum* with free access to water. Genetically modified mice used in this study include *Opn5^cre^*, *Ai14* (JAX stock 007914), and *Gnas^fl^* (JAX stock 035239) [32,33]. To account for paternal imprinting of *Gnas* [34], male *Gnas^fl/+^* and *Gnas^fl/fl^* mice were bred to female *Opn5^cre^; Gnas^fl/+^* mice to generate control male *Opn5^cre^; Gnas^+/+^* and experimental male *Opn5^cre^; Gnas^fl/fl^* mice, respectively. Littermate controls were used for all experiments. Only males were used for indirect calorimetry experiments.

### Lighting conditions

Mouse housing and metabolic experiments were on a 12:12 light:dark cycle. Mouse housing from birth until the time of experiment was in standard fluorescent lighting. Metabolic experiments were supplied with LED lighting that resembled standard fluorescent lighting. Data is presented as t = 0 when lights turn on at 6AM (Zeitgeber Time 0) and t = 12 when lights turn off at 6PM (ZT12).

### M-FISH

M-FISH experiments were performed as previously described [6]. Briefly, P42-49 male and female *Opn5^cre^;Gnas^fl/fl^;Ai14* (mutant) and *Opn5^cre^;Gnas^fl/+^;Ai14* or *Opn5^cre^;Gnas^+/+^;Ai14* (control) mice were euthanized by isoflurane and brains were rapidly dissected and snap-frozen in liquid nitrogen. 14-micron cryosections were obtained and processed using the RNA-scope Multiplex Fluorescent Reagent Kit v2 (ACD Bio). POA sections were fixed in ice-cold RNase-free paraformaldehyde for 1 hour and pre-treated in 1X RNase-free PBS and 50%, 75%, and 100% ethanol for five minutes each. After treatment with hydrogen peroxide for 15 minutes and protease for 30 minutes at 22˚C, sections were incubated in target probes against *Gnas* Exon 1 (Probe-Mm-Gnas-O1-C2, Cat No. 526891-C2) (1:50) and *tdTomato* (Probe-tdTomato-C3) (1:100) at 40˚C for 2 hours. Signal amplification using manufacturer-provided

Amp1, Amp2, and Amp3 reagents for 30, 30, and 15 minutes was performed at 40˚C, with 4-minute washes of 1X wash buffer in between. C2 and C3 were separately amplified with TSA Vivid Fluorophores 520 and 570 (1:1500 each), respectively. DAPI was stained for 15 seconds. Slides were then mounted using Tris-buffered Fluoro-Gel mounting medium (Electron Microscopy Sciences).

## M-FISH quantification

Brain sections were imaged using an AXR confocal microscope (Nikon). 40x z-stack fields of the POA were obtained. Each image contained DAPI for nuclear staining and signal from *Gnas* and *tdTomato* transcripts. FIJI (v1.53) was used for image processing and intensity analysis. Prior to processing, maximum intensity projections were generated and all channels were concatenated. Nuclei were detected and segmented using Otsu-LUT based thresholding and watershedding (BioVoxxel plugin), and individual masks were generated for each nucleus in the field. These masks were dilated by 1 pixel to capture intracellular signal outside the nuclear masks. Next, the average signal intensity for *Gnas* and *tdTomato* were extracted for each cell's mask and exported as a csv for further analysis in R (v4.0.0 –Arbor Day).

For cell masks, we computed the z-score of the log-normalized *Gnas* intensity for each image. This is necessary to eliminate slice-to-slice variation in RNA-Scope staining efficiency. This was repeated for *tdTomato* signal to subset masks that were positive for *tdTomato* (z-score $> 1.96$) for deletion efficiency analysis of *Gnas*. Z-score normalized intensity of *Gnas* in the *tdTomato* subset of cells was compared between WT (*Opn5$^{cre/+}$; Gnas$^{+/+}$; Ai14$^{+/-}$*), Het (*Opn5$^{cre/+}$; Gnas$^{fl/+}$; Ai14$^{+/-}$*), and KO (*Opn5$^{cre/+}$; Gnas$^{fl/fl}$; Ai14$^{+/-}$*) mice. Statistical comparisons, such as One-Way ANOVA with Tukey's HSD, were applied to means of animals (n = 2 WT, n = 2 Het, n = 3 KO) as opposed to cells (n = 63 WT, 61 Het, 114 KO).

## Indirect calorimetry and energy expenditure

6-to-12-week-old *Opn5$^{cre}$; Gnas$^{+/+}$* and *Opn5$^{cre}$; Gnas$^{fl/fl}$* mice were acclimated in Promethion Core metabolic chambers (Sable Systems International) one day before the start of the study. Mice were single-housed and continuously recorded with 15-minute measurements of the following parameters: oxygen consumption, carbon dioxide expiration, food intake, water intake, and locomotor activity. Ambient temperature was adjusted in a climate-controlled cabinet (**S1 Fig**). The two-day recording period at each temperature (**S1 Fig**, red rectangular outlines) was preceded by one day of same-temperature acclimation that was not included in analysis. Energy expenditure (EE) and respiratory exchange ratio (RER) were automatically calculated from vO$_2$ and vCO$_2$ measurements according to the manufacturer's guidelines. EE measurements were also analyzed with body mass. Food and water intake were recorded from top-fixed load cell sensors, with food and water suspended into the cage. Locomotion distance was measured by infrared XY beam breaks and includes both pedestrian and non-pedestrian motion. Sleep was measured as time of inactivity after 40 seconds of immobility, as a proportion of each 15-minute interval. All 48-hour measurements for each mouse were averaged into a 24-hour profile for analysis.

## Statistical analyses

Statistical analysis was performed using GraphPad Prism version 9.3.1 (GraphPad Software), Microsoft Excel, MATLAB, R 4.0.0, and FIJI (v1.53). For M-FISH, one-way ANOVA with Tukey's HSD was used to compare genotypes: WT (*Opn5$^{cre}$;Gnas$^{+/+}$; Ai14$^{+/-}$*, n = 2*)*, Het (*Opn5$^{cre/+}$; Gnas$^{fl/+}$; Ai14$^{+/-}$*, n = 2), and KO (*Opn5$^{cre/+}$; Gnas$^{fl/fl}$; Ai14$^{+/-}$*, n = 3). 48-hour measurements were averaged into 24 hours for each mouse. Two-way ANOVA with Sidak's

multiple comparison's tests were used for each measurement within 3-hour bins of average values. Energy-mass relationships were calculated by ANCOVA with body mass as a covariate. Differential rhythmic parameters including amplitude, phase, and MESOR differences were analyzed using the CircaCompare R package [35]. Differences were considered significant if $p < 0.05$.

# Results

## *Gnas* deletion in QPLOT neurons

GNAS activation is known to increase cAMP through adenylyl cyclase, and OPN5 stimulation in preoptic area slices increases cAMP [6]. This provided a strong rationale for an investigation of GNAS function in QPLOT neurons. We conditionally deleted the *Gnas*[fl] allele [33] in QPLOT neurons by crossing *Gnas*[fl/fl] mice to mice expressing the *Opn5*[cre] driver [32]. Deletion of Exon 1 in *Gnas*[fl] by *cre* recombinase is expected to eliminate all $G_{\alpha s}$ function [33]; wild-type *Gnas* is unaffected by *cre*. *Opn5* is among the most conserved and specific markers of QPLOT neurons [23] and so represents an excellent option for the investigation of QPLOT neurons. *Opn5*-null mice show increased thermogenesis because of an inability of *Opn5* POA neurons to suppress SNS output. Designer-receptors-exclusively-activated-by-designer-drugs (DREADDs) expressed by *Opn5*[cre] in the POA further confirm the thermoregulatory role of *Opn5* neurons; stimulation or inhibition of these neurons immediately decreases or increases EE, respectively, which directly adjusts body temperature [6]. Together, these observations support the use of *Opn5*[cre] as a tool to understand $G_{\alpha s}$-signaling in the QPLOT neuron population.

Consistent with published immunofluorescence data [6], *Opn5*[cre] is expressed only in the hypothalamic preoptic area (**Fig 1B**). To assess deletion of *Gnas* in *Opn5* neurons, we used *in situ* hybridization to detect *Gnas* Exon 1 transcript in *Opn5*[cre]; *Gnas*[+/+], *Opn5*[cre];*Gnas*[fl/+], and *Opn5*[cre];*Gnas*[fl/fl] brain sections using *cre*-dependent *Ai14* (tdTomato) expression to label *Opn5*[+] cells. Compared to *Opn5*[cre]; *Gnas*[+/+];*Ai14* cells, we detected substantially reduced levels of *Gnas* transcript in both *Opn5*[cre]; *Gnas*[fl/+]; *Ai14* and *Opn5*[cre]; *Gnas*[fl/fl]; *Ai14* neurons (**Fig 1C and 1D**), which suggests that *Opn5*[cre] inefficiently deletes both copies of *Gnas*. Therefore, for metabolic experiments, we used only *Opn5*[cre];*Gnas*[+/+] and *Opn5*[cre];*Gnas*[fl/fl] mice.

## Measurement of energy expenditure

To assess how the loss of GNAS function in QPLOT neurons alters mouse metabolism, we generated cohorts of control (*Opn5*[cre]; *Gnas*[+/+]) and experimental (*Opn5*[cre]; *Gnas*[fl/fl]) mice and analyzed them using an indirect calorimetry cage system. The metabolic performance of mice was analyzed over a twelve-day protocol in which three days of housing at 22°C (standard ambient temperature) was ramped down over two days to a cold challenge of 10°C, maintained for three days and then ramped up over one day to three days at 28°C (approximately thermoneutrality, the temperature at which basal metabolic rate is sufficient to maintain body temperature) (**S1 Fig**). This protocol gave the mice one day to acclimate after each temperature shift. Because many metabolic parameters follow 24-hour circadian rhythms, we performed analysis through CircaCompare [35]. This statistical method generates regression-based cosinor curves and compares central rhythmic values such as MESOR (Midline Estimating Statistic of Rhythm, or rhythm-adjusted mean value), amplitude (half the distance between peak and trough), and phase (measured by time of peak value). We also analyzed the data using a conventional two-way ANOVA. As an overview of the analysis, we summarized the distribution of statistically significant changes between *Opn5*[cre]; *Gnas*[+/+] and *Opn5*[cre]; *Gnas*[fl/fl] mice

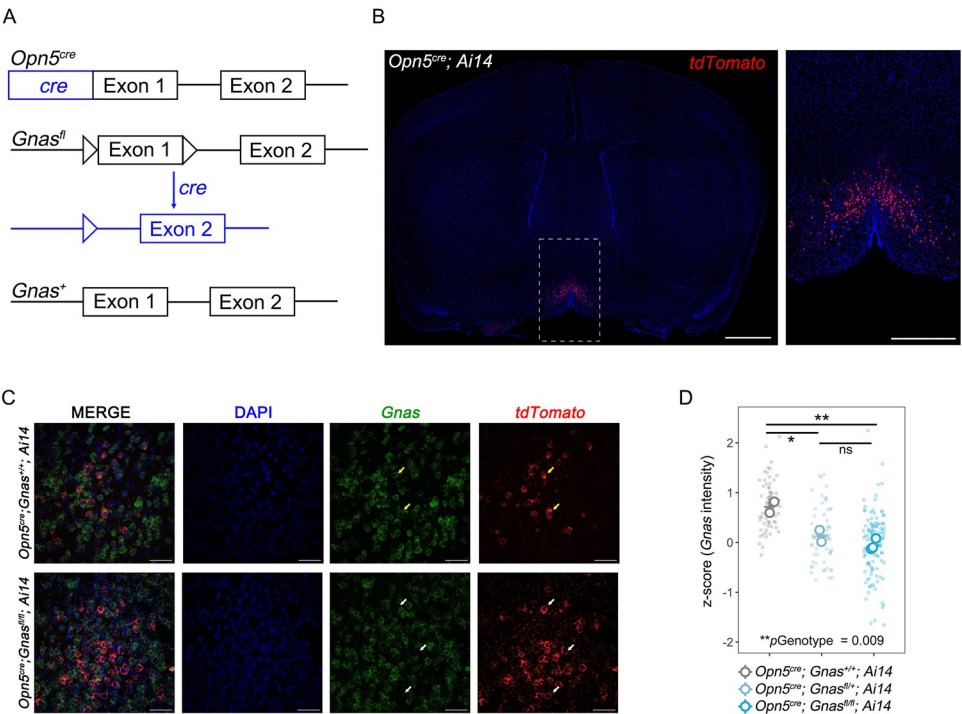

**Fig 1. *Gnas* deletion in hypothalamic *Opn5* neurons.** A. Schematic of conditional *Gnas* deletion in *Opn5* cells. Recombination of loxP sites by *Opn5^cre^* excises *Gnas* Exon 1 in the conditional allele (*Gnas^fl^*). The wild-type *Gnas* allele (*Gnas^+^*) does not have loxP sites and is unaffected by *cre*. B. Representative coronal brain section of *Opn5^cre^; Ai14* mice (scale bar, 1mm). *Ai14* (*tdTomato*, red) expression by *Opn5^cre^* is only in the hypothalamic preoptic area (inset; scale bar, 500 μm). C. Representative images showing co-localization of *Gnas* (green) in *tdTomato^+^* (red) cells in control (*Opn5^cre^;Gnas^+/+^;Ai14*, top row) and mutant (*Opn5^cre^;Gnas^fl/fl^;Ai14*, bottom row) coronal brain sections. Yellow arrows indicate co-localization of *Gnas* and *tdTomato*; white arrows show no co-localization. Scale bars, 50 μm. D. Quantification of z-scores of *Gnas* intensity in *Ai14^+^* cells in *Opn5^cre/+^; Gnas^+/+^; Ai14^+/-^*, (n = 2 mice, gray), *Opn5^cre/+^; Gnas^fl/+^; Ai14^+/-^* (n = 2 mice, light blue), and *Opn5^cre/+^; Gnas^fl/fl^; Ai14^+/-^* (n = 3 mice, blue). Data is presented as mean± s.e.m. and analyzed by one-way ANOVA with Tukey's HSD. *p<0.05; **p<0.01.

over the daily time-course at different temperatures (**Tables 1 and 2**). This helps to establish the overall pattern of change.

Energy expenditure (EE) is calculated from measurements of oxygen consumption and carbon dioxide expiration using the Weir formula [36]. According to an assessment of statistical significance by ANOVA, *Opn5^cre^; Gnas^fl/fl^* mice show no significant differences in EE at any temperature or time of day (**Fig 2A–2C**). Because body mass influences EE, we corrected metabolic rate by mass by performing ANCOVA on the energy measurements at each temperature with body mass as a covariate. At all temperatures, experimental and control mice showed no overall difference in energy-mass relationships (**Fig 2D–2F**).

However, when we applied the CircaCompare method to assess statistical significance in data with circadian rhythmicity, several differences were identified. As expected, CircaCompare showed that EE in both genotypes was significantly rhythmic with a period estimate of 24 hours at all temperatures (**Fig 2G–2L**), recapitulating natural variations with a light-dark cycle. The MESOR for the EE curve was not different when comparing control and *Opn5^cre^; Gnas^fl/fl^* mice at 28˚C (**Fig 2G and 2J**) but was significantly higher in *Opn5^cre^; Gnas^fl/fl^* mice at both 22˚C (**Fig 2H** and **2K**) and 10˚C (**Fig 2I** and **2L**). Furthermore, we identified both a minor amplitude decrease and a 0.64-hour phase delay in the diurnal rhythmicity of EE for *Opn5^cre^; Gnas^fl/fl^* mice at 28˚C (**Fig 2G** and **2J**). Overall, these data suggest that the circadian pattern of energy expenditure is subtly regulated by the activity of $G_{\alpha s}$ signaling in QPLOT neurons.

**Table 1. Summary of metabolic analysis of *Opn5^cre^; Gnas^+/+^* and *Opn5^cre^; Gnas^fl/fl^* mice by two-way ANOVA.**

| Thermoneutral (28°C) | Overall | 0–3 hrs | 3–6 hrs | 6–9 hrs | 9–12 hrs | 12–15 hrs | 15–18 hrs | 18–21 hrs | 21–24 hrs |
|---|---|---|---|---|---|---|---|---|---|
| EE | | | | | | | | | |
| RER | | | | | | | | | |
| Food Intake | | | | | | | | | |
| Water Intake | | | | | | | | | |
| All Locomotion | Int: p < .0001; Geno: p = .0385 | | | | | Control | | | Control |
| **Standard Ambient (22°C)** | **Overall** | **0–3 hrs** | **3–6 hrs** | **6–9 hrs** | **9–12 hrs** | **12–15 hrs** | **15–18 hrs** | **18–21 hrs** | **21–24 hrs** |
| EE | | | | | | | | | |
| RER | | | | | | | | | |
| Food Intake | | | | | | | | | |
| Water Intake | | | | | | | | | |
| All Locomotion | Int: p = .0023 | | | | | Control | | | |
| **Cold-Stress (10°C)** | **Overall** | **0–3 hrs** | **3–6 hrs** | **6–9 hrs** | **9–12 hrs** | **12–15 hrs** | **15–18 hrs** | **18–21 hrs** | **21–24 hrs** |
| EE | | | | | | | | | |
| RER | | | | | | | | | |
| Food Intake | | | | | | | | | |
| Water Intake | | | | | | | | | |
| All Locomotion | | | | | | | | | |

EE, energy expenditure; RER, respiratory exchange ratio; Int, Interaction; Geno, Genotype.

[a]p-values were calculated using two-way ANOVA.

[b]Gray boxes indicate that the control value was higher than the mutant value; blue boxes indicate that the mutant value was higher than the control value.

**Table 2. Summary of metabolic analysis of *Opn5^cre^; Gnas^+/+^* and *Opn5^cre^; Gnas^fl/fl^* mice by CircaCompare.**

| Thermoneutral (28°C) | Rhythmicity | MESOR | Amplitude | Phase |
|---|---|---|---|---|
| EE | Yes | | Low | Delayed 0.64h |
| RER | Yes | High | High | Delayed 1.16h |
| Food Intake | Yes | High | | |
| Water Intake | Yes | High | | |
| All Locomotion | Yes | Low | Low | |
| **Standard Ambient (22°C)** | **Rhythmicity** | **MESOR** | **Amplitude** | **Phase** |
| EE | Yes | High | | |
| RER | Yes | Low | High | |
| Food Intake | Yes | High | High | |
| Water Intake | Yes | | | |
| All Locomotion | Yes | Low | Low | |
| **Cold-Stress (10°C)** | **Rhythmicity** | **MESOR** | **Amplitude** | **Phase** |
| EE | Yes | High | | |
| RER | Yes | Low | | |
| Food Intake | Yes | | | |
| Water Intake | Yes | | | |
| All Locomotion | Yes | Low | | |

EE, energy expenditure; RER, respiratory exchange ratio.

[a]p-values were calculated using CircaCompare.

[b]Gray boxes indicate that the mutant value was lower than the control value; blue boxes indicate that the mutant value was higher than the control value.

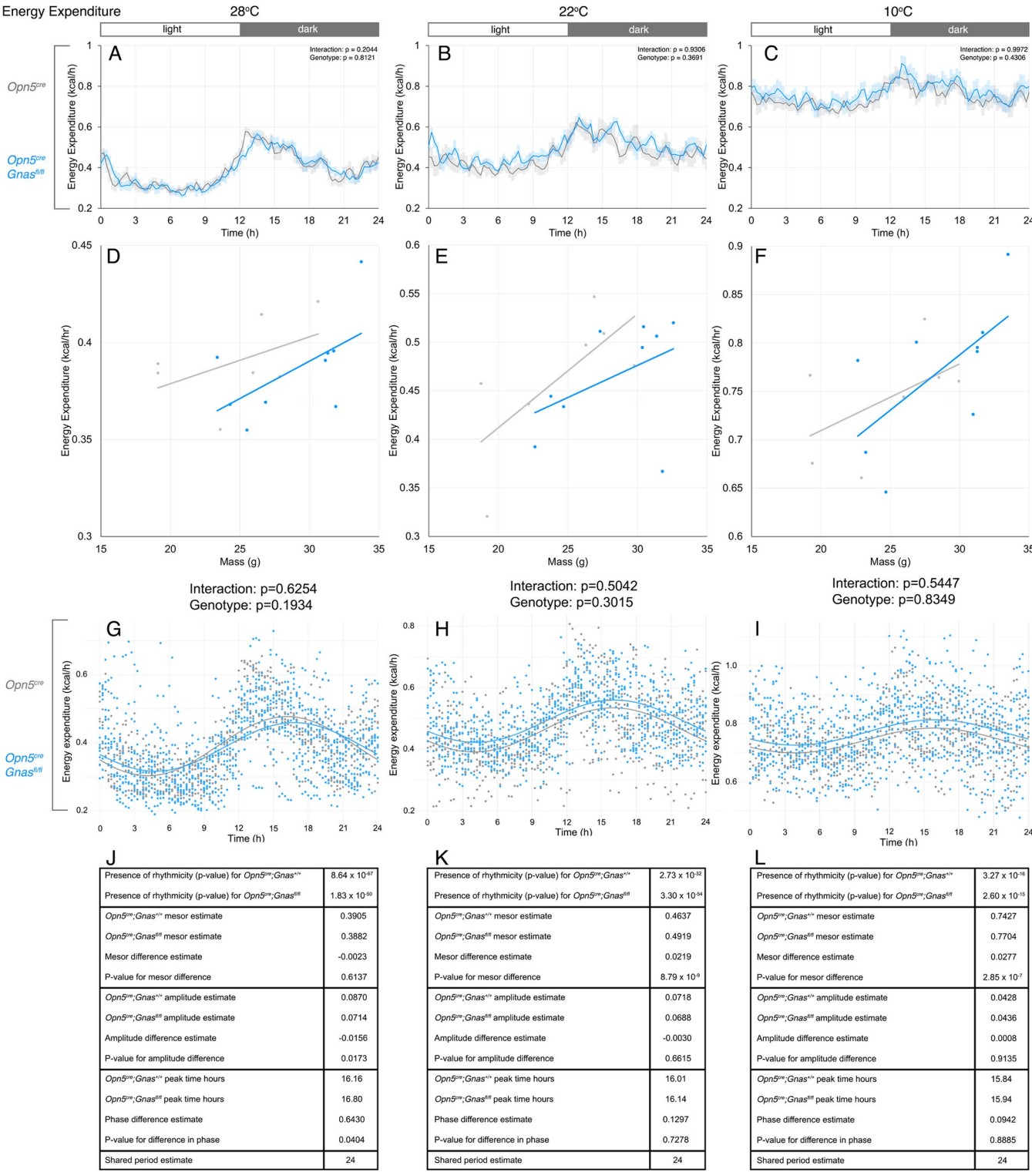

**Fig 2. Daily energy expenditure of *Gnas* conditional mutants and controls at different temperatures.** Energy expenditure in *Opn5^cre^; Gnas^fl/fl^* (n = 9, blue) and *Opn5^cre^; Gnas^+/+^* (n = 7, gray) mice was recorded by indirect calorimetry. Respective temperatures are indicated at the top of each column. For time of day on x-axes, t = 0 represents the start of the light phase at 6AM and t = 12 represents the start of the dark phase at 6PM (light:dark bar above each column). 24-hour energy expenditure profiles (A-C) at thermoneutrality (A), standard ambient temperature (B), and during cold-stress (C). Data points in A-C are presented as mean± s.e.m. for 15-minute measurements and analyzed by two-way ANOVA for each 3-hour interval. Significant p-values are indicated above each interval. Energy-mass relationships (D-F) from 24-hour data in A-C were analyzed by ANCOVA at each temperature with mass as the covariate. P-values

of genotype and genotype x mass interaction for D-F are presented below each graph. Cosinor curves (G-I) were generated by CircaCompare for thermoneutrality (G), standard ambient temperature (H), and during cold-stress (I). Estimates of circadian values and p-values (Tables J-L) are listed under each cosinor curve.

## Respiratory exchange

The respiratory exchange ratio (RER) is calculated as the ratio between carbon dioxide production and oxygen consumption and reflects an animal's substrate preference. The RER value varies between 1.0 for carbohydrate metabolism, observed after eating, and decreases to about 0.7 when lipid metabolism is preferred, such as during periods of fasting [37]. RER may exceed 1 during intense physical activity [38] or lipogenesis [39]. RER is also expected to decrease at lower ambient temperatures, as a higher energy demand to maintain body temperature would recruit lipid oxidation [40].

Despite an apparent divergence in RER values during the 28˚C dark phase (**Fig 3A**), we observed no statistical difference between the two genotypes at any temperature according to two-way ANOVA (**Fig 3A–3C**). However, we observed a striking temperature-dependent shift in RER rhythms using CircaCompare. At all temperatures, RER is significantly rhythmic with a period estimate of 24 hours (**Fig 3D–3I**). At 28˚C, the MESOR of *Opn5^cre; Gnas^fl/fl* mice is higher than that of control mice (**Fig 3D** and **3G**). Interestingly, the MESOR of *Opn5^cre; Gnas^fl/fl* mice shifts below that of controls at 22˚C (**Fig 3E** and **3H**), and remains lower than the control MESOR at 10˚C (**Fig 3F** and **3I**). Furthermore, CircaCompare analysis showed that the amplitude of RER is significantly increased in *Opn5^cre; Gnas^fl/fl* mice at 28˚C (**Fig 3D** and **3G**) but is decreased in experimental mice at 22˚C (**Fig 3E** and **3H**). Finally, similar to our findings for EE at 28˚C, CircaCompare detected a phase delay in RER of about 1.16 hours (**Fig 3D** and **3G**). These data suggest a temperature-dependent shift in the substrate preference of *Opn5^cre; Gnas^fl/fl* mice as a consequence of the loss of G_αs-signaling in QPLOT neurons.

## Food and water intake

In addition to metabolic rate, animals may display behavioral adaptations as a component of energy homeostasis. For example, food and water intake display diurnal rhythms. Mice show the highest consumption occurring early in the dark phase, and animals may change food and water intake in response to changes in ambient temperature [22]. Thus, we were interested in determining if G_αs-signaling in QPLOT neurons regulates food and water consumption. We used two-way ANOVA to assess food intake between *Opn5^cre; Gnas^fl/fl* and control mice across temperatures, and we observed no significant changes at any temperature or time of day (**Fig 4A–4C**). However, we observed differences in the rhythm of food intake at the higher temperatures using CircaCompare. At all temperatures, *Opn5^cre; Gnas^fl/fl* mice and control mice showed significantly rhythmic patterns of food intake with a period estimate of 24 hours (**Fig 4D–4I**). At 28˚C, *Opn5^cre; Gnas^fl/fl* mice showed a high MESOR in food intake, but not amplitude or phase difference (**Fig 4D and 4G**). At 22˚C, *Opn5^cre; Gnas^fl/fl* mice showed both a high MESOR and amplitude in food intake compared with *Opn5^cre; Gnas^+/+* mice (**Fig 4E and 4H**). Finally, there were no changes in food intake at 10˚C (**Fig 4F and 4I**). These data indicate that G_αs-signaling in QPLOT neurons subtly influences the circadian pattern of food consumption at standard ambient temperature and above.

Water intake may also shift in response to temperature shifts. We did not observe any changes in water consumption between *Opn5^cre; Gnas^fl/fl* mice and control mice at any temperature using two-way ANOVA (**Fig 5A–5C**). However, we detected a subtle shift in water intake using CircaCompare analysis. Like food intake, the circadian pattern of water intake was

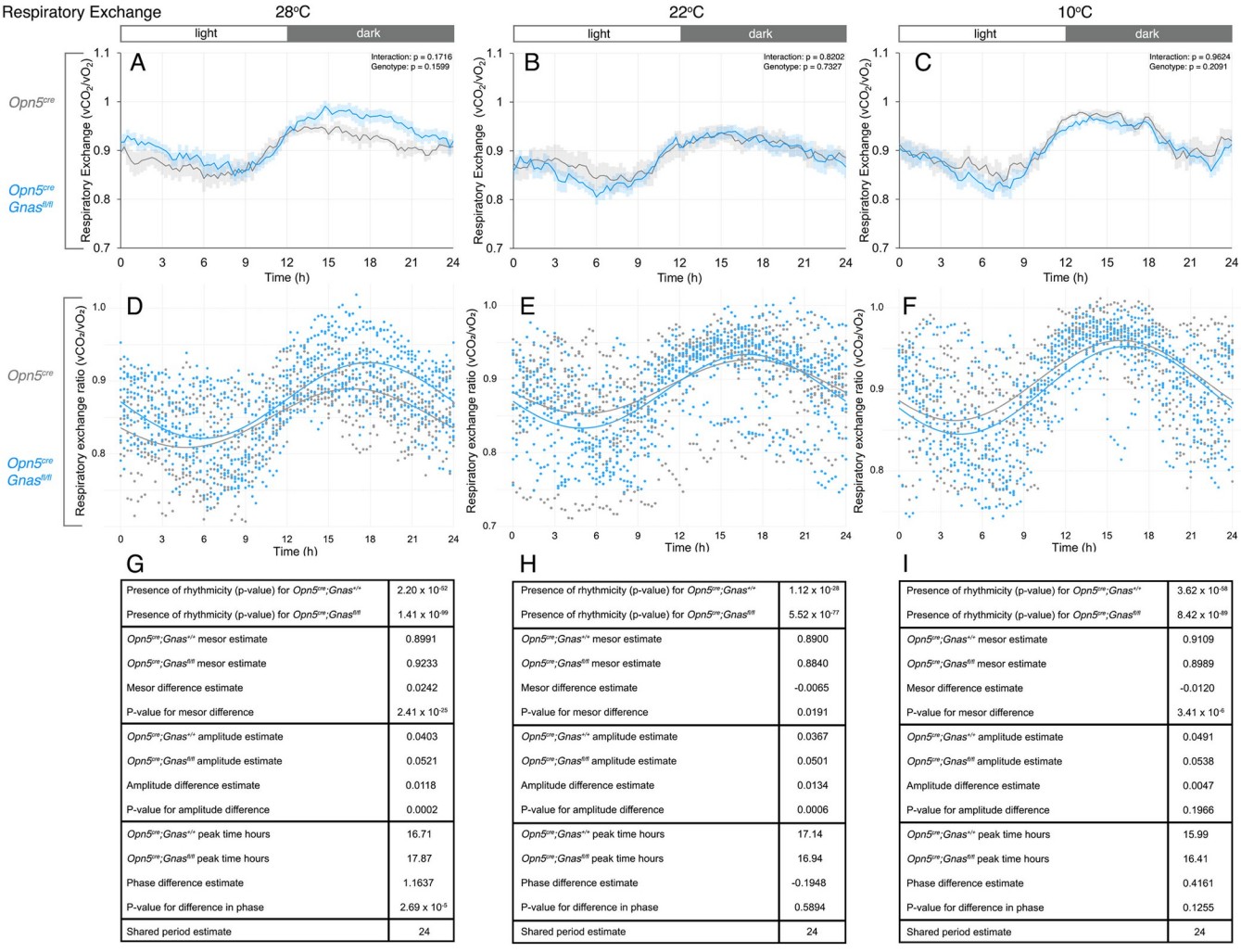

**Fig 3. Daily respiratory exchange ratios of *Gnas* conditional mutants and controls at different temperatures.** Respiratory exchange ($vCO_2/vO_2$) in *Opn5^cre^; Gnas^fl/fl^* (n = 9, blue) and *Opn5^cre^; Gnas^+/+^* (n = 7, gray) mice was recorded by indirect calorimetry. Respective temperatures are indicated at the top of each column. For time of day on x-axes, t = 0 represents the start of the light phase at 6AM and t = 12 represents the start of the dark phase at 6PM (light:dark bar above each column). 24-hour respiratory exchange ratios (A-C) at thermoneutrality (A), standard ambient temperature (B), and during cold-stress (C). Data points in A-C are presented as mean± s.e.m. for 15-minute measurements and analyzed by two-way ANOVA for each 3-hour interval. Significant p-values are indicated above each interval. Cosinor curves (D-F) were generated by CircaCompare for thermoneutrality (D), standard ambient temperature (E), and during cold-stress (F). Estimates of circadian values and p-values (G-I) are listed under each cosinor curve.

significantly rhythmic with a period estimate of 24 hours (**Fig 5D–5I**). At 28˚C, we detected an increased MESOR for water intake in *Opn5^cre^; Gnas^fl/fl^* mice, but there were no changes in amplitude or phase (**Fig 5D and 5G**). However, at 22˚C and 10˚C, water intake was no different between control and experimental mice (**Fig 5E, 5F, 5H and 5I**). These differences suggest that $G_{\alpha s}$-signaling in QPLOT neurons may regulate the circadian pattern of water intake at higher temperatures.

## Locomotion

In nocturnal species such as mice, physical activity follows a circadian pattern in which locomotion increases early in the dark phase but decreases throughout the night and stays low during the light phase. Locomotion represents an additional behavioral response to temperature

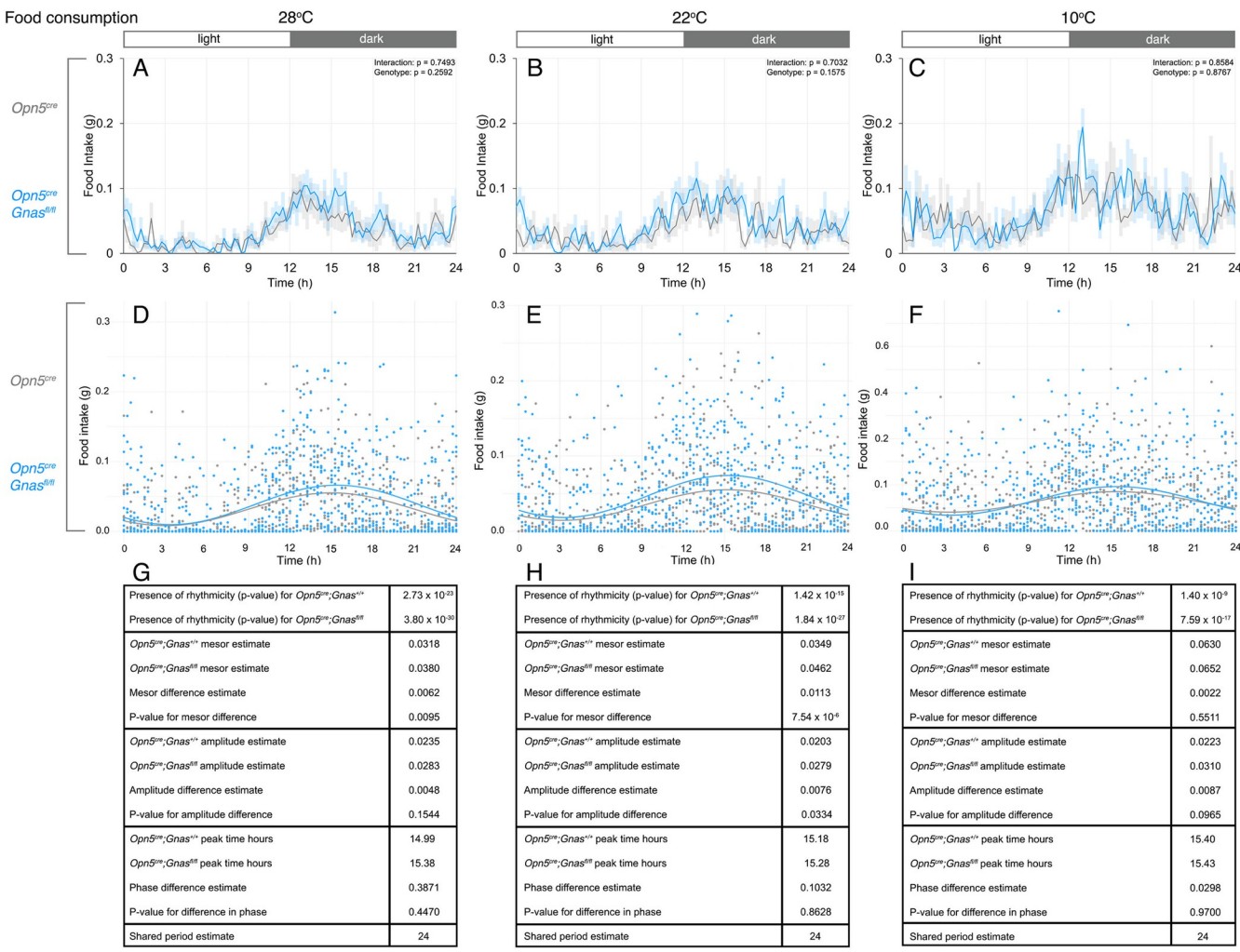

**Fig 4. Daily food intake of *Gnas* conditional mutants and controls at different temperatures.** Food intake in *Opn5^cre^; Gnas^fl/fl^* (n = 9, blue) and *Opn5^cre^; Gnas^+/+^* (n = 7, gray) mice was recorded by weight-based sensors. Respective temperatures are indicated at the top of each column. For time of day on x-axes, t = 0 represents the start of the light phase at 6AM and t = 12 represents the start of the dark phase at 6PM (light:dark bar above each column). 24-hour food consumption profiles (A-C) at thermoneutrality (A), standard ambient temperature (B), and during cold-stress (C). Data points in A-C are presented as mean± s.e.m. for 15-minute measurements and analyzed by two-way ANOVA for each 3-hour interval. Significant p-values are indicated above each interval. Cosinor curves (D-F) were generated by CircaCompare for thermoneutrality (D), standard ambient temperature (E), and during cold-stress (F). Estimates of circadian values and p-values (G-I) are listed under each cosinor curve.

changes, as physical activity generates heat [11]. Using two-way ANOVA, we observed a striking decrease in locomotion in *Opn5^cre^; Gnas^fl/fl^* mice at 28˚C and 22˚C (**Fig 6A and 6B**). This significant decrease is attributable to a noticeable reduction in distance travelled in the early night at 28˚C and 22˚C, which is when locomotion is expected to increase in nocturnal animals (**Fig 6A and 6B**). Locomotion was also decreased in *Opn5^cre^; Gnas^fl/fl^* mice at the end of the dark phase at 28˚C.

We also used CircaCompare to evaluate the rhythmic characteristics of the daily locomotion patterns of *Opn5^cre^; Gnas^fl/fl^* and control mice. At all temperatures, as expected, we observed significant rhythmicity in locomotor patterns for both *Opn5^cre^; Gnas^fl/fl^* and control mice, with a period estimate of 24 hours (**Fig 6D–6I**). Interestingly, we also observed a low MESOR in locomotion in *Opn5^cre^; Gnas^fl/fl^* mice at all temperatures (**Fig 6D–6I**). At 28˚C and 22˚C, CircaCompare analysis also revealed a low amplitude in locomotion in *Opn5^cre^; Gnas^fl/fl^*

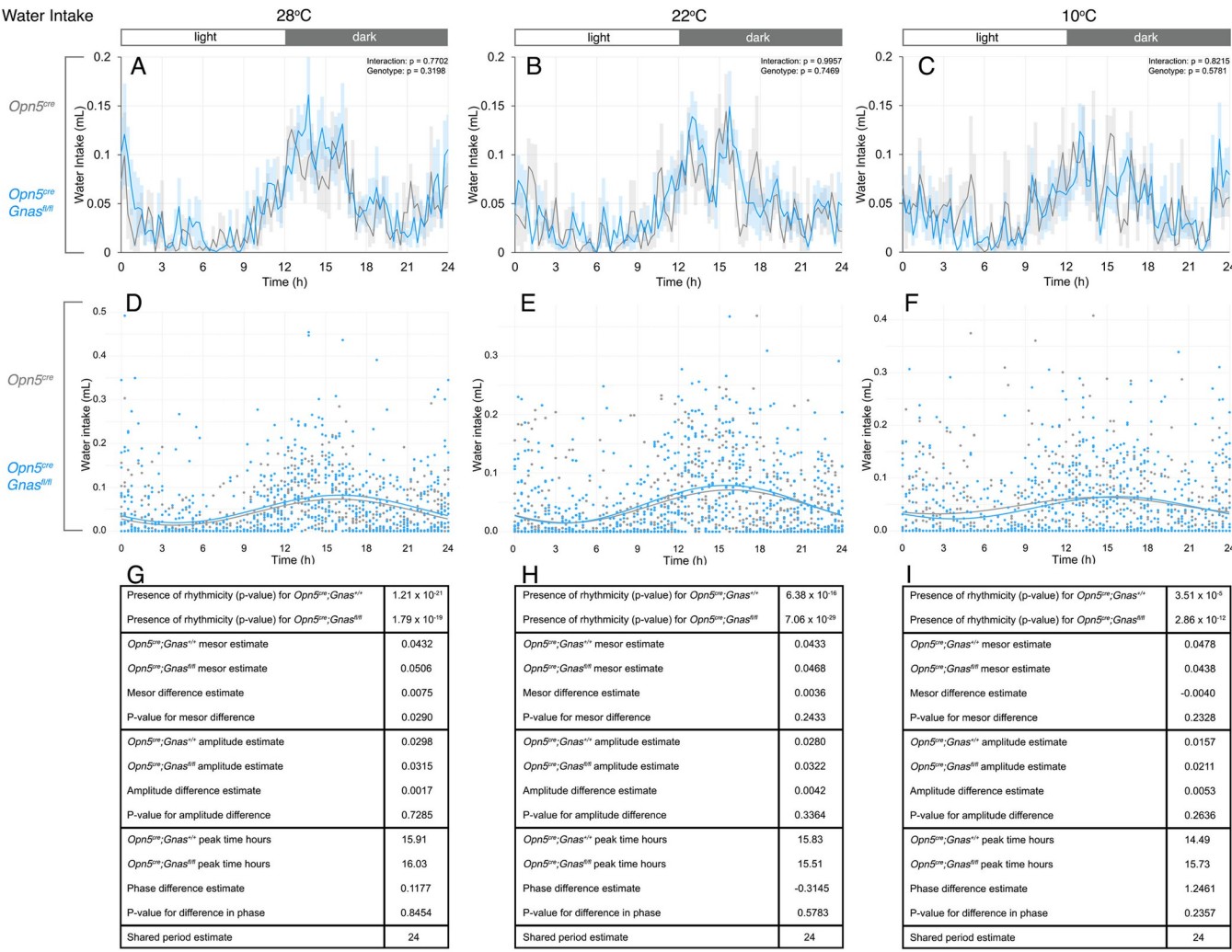

**Fig 5. Daily water intake of *Gnas* conditional mutants and controls at different temperatures.** Water intake in *Opn5<sup>cre</sup>; Gnas<sup>fl/fl</sup>* (n = 9, blue) and *Opn5<sup>cre</sup>;* *Gnas<sup>+/+</sup>* (n = 6, gray) mice was recorded by weight-based sensors. Respective temperatures are indicated at the top of each column. For time of day on x-axes, t = 0 represents the start of the light phase at 6AM and t = 12 represents the start of the dark phase at 6PM (light:dark bar above each column). 24-hour water consumption profiles (A-C) at thermoneutrality (A), standard ambient temperature (B), and during cold-stress (C). Data points in A-C are presented as mean± s.e.m. for 15-minute measurements and analyzed by two-way ANOVA for each 3-hour interval. Significant p-values are indicated above each interval. Cosinor curves (D-F) were generated by CircaCompare for thermoneutrality (D), standard ambient temperature (E), and during cold-stress (F). Estimates of circadian values and p-values (G-I) are listed under each cosinor curve.

mice, which is consistent with the nocturnal locomotor decrease detected by ANOVA (**Fig 6A, 6B, 6D, 6E 6G and 6H**). Finally, no differences in phase were observed at any temperature (**Fig 6D, 6E, 6F, 6G, 6H and 6I**). These results suggest a potential role for G<sub>αs</sub>-signaling in QPLOT neurons in the regulation of physical activity, with the strongest influence occurring at standard ambient temperature and higher.

In conjunction with locomotion, sleep was measured using extended immobility as a proxy (**S2 Fig**). We detected no overall changes in sleep across temperatures (S2A-S2C Fig) as measured by the duration of inactivity. Furthermore, rhythmic analysis of sleep duration revealed no changes in the daily rhythm of sleep (**S2D-S2I Fig**) across temperatures. Therefore, while *Opn5<sup>cre</sup>;Gnas<sup>fl/fl</sup>* mice demonstrate no change in the time or rhythm of sleep, they commute less distance during the dark phase.

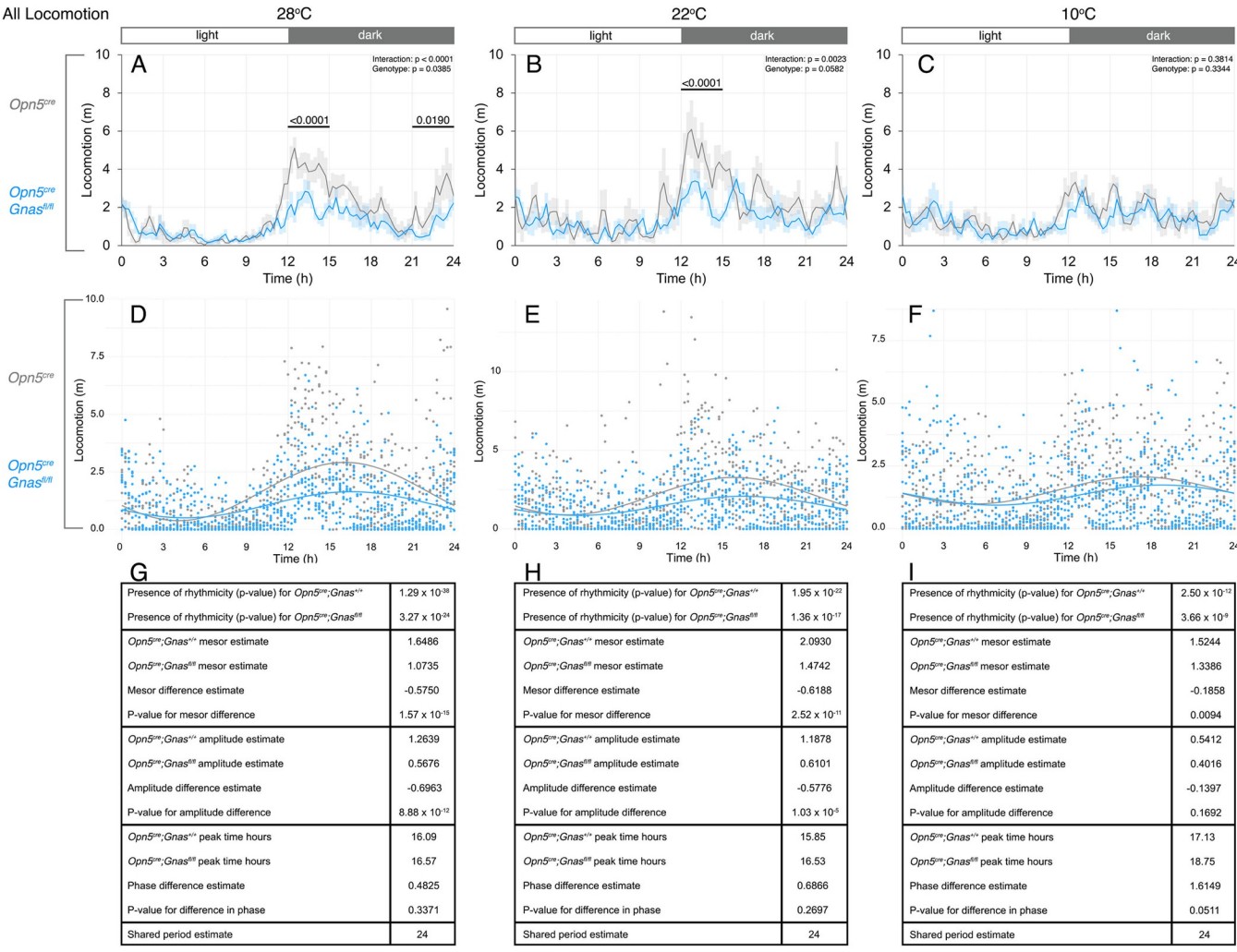

**Fig 6. Daily locomotor activity of *Gnas* conditional mutants and controls at different temperatures.** Locomotor distance traveled in *Opn5^{cre}; Gnas^{fl/fl}* (n = 9, blue) and *Opn5^{cre}; Gnas^{+/+}* (n = 7, gray) mice was passively recorded by infrared beam breaks. Respective temperatures are indicated at the top of each column. For time of day on x-axes, t = 0 represents the start of the light phase at 6AM and t = 12 represents the start of the dark phase at 6PM (light:dark bar above each column). 24-hour locomotor profiles (A-C) at thermoneutrality (A), standard ambient temperature (B), and during cold-stress (C). Data points in A-C are presented as mean± s.e.m. for 15-minute measurements and analyzed by two-way ANOVA for each 3-hour interval. Significant p-values are indicated above each interval. Cosinor curves (D-F) were generated by CircaCompare for thermoneutrality (D), standard ambient temperature (E), and during cold-stress (F). Estimates of circadian values and p-values (G-I) are listed under each cosinor curve.

## Discussion

In this study, we assessed the metabolic effects of deleting GNAS/G$_{\alpha s}$ signaling in QPLOT neurons. QPLOT neurons act to reduce core body temperature and metabolism when activated, so removing a stimulatory mechanism such as G$_{\alpha s}$-signaling is hypothesized to prevent their activation and increase metabolism.

We used *Opn5^{cre}* to delete *Gnas* because *Opn5* is a highly conserved marker in the QPLOT population [23]. In the adult central nervous system, *Opn5^{cre}* is only detected in the hypothalamic preoptic area and a subset of retinal ganglion cells (RGCs) [6,32,41]. *Opn5* mutant mice experience elevated metabolism secondary to increased SNS activity [6] and this is attributed to the *Opn5*-expressing neurons in the hypothalamic preoptic area. Enucleated mice, which experience no influence of retinal *Opn5*, show similar metabolic perturbations as global *Opn5*

mutants [6]. Retinal *Opn5* is confined to a small subset of all RGCs, and these *Opn5*-expressing RGCs do not project to the SCN [41]. Therefore, it is very unlikely that the daily rhythmic differences we observed can be attributed to retinal *Opn5* cells. The *Opn5*$^{cre}$ allele is a knock-in design and equivalent to *Opn5* heterozygosity. We detected a low level of *Gnas* signal in our *Opn5*$^{cre}$; *Gnas*$^{fl/fl}$ mice, which suggests imperfect deletion in *Opn5* neurons and may contribute to the relatively mild phenotypes we observed. Analysis of *Opn5* heterozygotes for changes in metabolic responses has not shown haploinsufficiency [6].

The $G_{\alpha s}$ subunit of the trimeric G-protein complex normally stimulates cAMP production through activation of transmembrane adenylyl cyclases. $G_{\alpha s}$ does not regulate soluble adenylyl cyclase, which is regulated by intracellular ions and not directly by GPCRs [42]. Therefore, the expectation is that *cre*-mediated deletion of *Gnas* would eliminate all cAMP signaling through GPCRs, but that some GPCR-independent cAMP generation by soluble adenylyl cyclase would remain. A rescue experiment in which *Gnas* is reintroduced to QPLOT neurons would confirm that $G_s$-signaling plays a role in maintaining daily rhythmic cycles. Additionally, activation of $G_s$-coupled DREADDs [43] in QPLOT neurons may influence daily rhythms. Furthermore, $G_s$-signaling is only one of several signaling pathways in QPLOT neurons that could regulate metabolism. Because QPLOT neurons can exert profound effects on metabolism and express several important receptors that are GPCRs (EP3, OPN5, NK3R) or non-GPCRs (LEPR), these neurons must be well-regulated so that a stimulus can produce an appropriate adaptation without wasting energy. Therefore, the interplay of multiple intracellular signaling pathways, including $G_s$, may provide resilience to QPLOT neuron activity and resist major disturbances to a single pathway.

We observed several rhythmic shifts in the metabolism of *Opn5*$^{cre/+}$; *Gnas*$^{fl/fl}$ mutant mice across temperatures. Using ANOVA to assess all parameters, we only observed a striking difference in locomotion, in which the distance traveled in the first hours of the night was decreased in the *Gnas* mutants at 28˚C and 22˚C, without concomitant changes in the other parameters, including apparent sleep duration. The activity of nocturnal species such as mice increases sharply around the start of the dark phase and gradually decreases throughout the night, with expected concurrent increases in food intake and energy expenditure [11,44]. While QPLOT neurons have been associated with control of adaptive thermogenesis in BAT, they have not been directly associated with locomotor activity. Therefore, the cause of decreased nocturnal locomotion distance is yet to be determined. Concomitant with this difference, we observed reductions in the MESOR and amplitude of locomotion at 28˚C and 22˚C using CircaCompare, which were consistent with the analysis with ANOVA. The activity of $G_{\alpha s}$ signaling in QPLOT neurons may directly influence locomotion, or it may regulate other energy use such that locomotion is secondarily affected.

EE is determined by locomotion, food intake, cold-induced thermogenesis, and basal metabolic rate. At 28˚C, we did not observe any differences in EE using ANOVA, but CircaCompare showed that EE was phase-delayed by about 0.64 hours. This phase-delay suggests that loss of $G_{\alpha s}$ signaling interferes with normal EE rhythms. However, we observed an increased MESOR in EE at 22˚C and 10˚C. These temperatures are below thermoneutrality and would be expected to trigger adaptive thermogenic responses. QPLOT neurons are warm-sensing [21,23] and may use $G_{\alpha s}$ signaling for activation in response to heat stimuli, suppressing thermogenic processes. This mechanism may explain the phase-delay in EE at 28˚C and increased MESOR at lower temperatures. If $G_{\alpha s}$-signaling is a major pathway for warm-responsive behaviors, deletion of $G_{\alpha s}$-signaling may delay warm responses until alternate pathways can be activated. Additional analysis would be required to determine the components of EE that are most significantly affected.

Furthermore, we observed a temperature-dependent change in RER rhythms in *Gnas* conditional mutants. In general, RER is highest during the active phase, when food intake and

locomotor activity are highest, and carbohydrate consumption satisfies energy requirements [45]. Conversely, RER decreases during the inactive phase, when mice are fasting and fatty acids from adipose tissue provide energy [46]. RER also decreases at low temperatures when adaptive thermogenesis recruits white adipose tissue through the SNS [47,48]. Two-way ANOVA did not identify any significant differences in RER. However, CircaCompare revealed significantly modified RER rhythms with decreasing temperature. While the RER MESOR is higher in *Gnas* mutants at 28˚C, it drops below that of control mice at 22˚C and 10˚C. This pattern indicates that carbohydrate utilization is higher in *Gnas* mutants at 28˚C, when carbohydrate use is expected to be relatively abundant, but lipid consumption is more preferred than in controls at lower temperatures. When considered in context with elevated EE MESOR at lower temperatures, *Gnas* mutant mice consume more energy at 22˚C and 10˚C, likely due to increased lipid use compared to control mice. Furthermore, the RER rhythm is phase-delayed by 1.16 hours in *Gnas* mutants at 28˚C. At this temperature, EE is also delayed, suggesting that *Gnas* mutants may initially be slow to recruit carbohydrate sources, but later surpass the expected extent of carbohydrate consumption.

$G_{\alpha s}$ drives intracellular signaling upon stimulation of a $G_{\alpha s}$-coupled receptor. The receptors responsible for our observed metabolic differences are yet to be determined. While OPN5 is classified as a $G_i$-coupled opsin *in vitro* [49], acute stimulation by violet light of *Opn5* neurons in *ex vivo* brain slices increases cAMP, perhaps suggesting a role for $G_{\alpha s}$-signaling during *Opn5* activity [6]. OPN5 has also been connected to $G_q$-signaling *in vivo* and *in vitro*, which suggests OPN5 may couple to multiple G-protein subtypes [50]. Similar temperature stresses have been conducted on *Opn5$^{-/-}$* mice. *Opn5* homozygous loss-of-function mice show elevated metabolism by ANOVA at lower temperatures even without OPN5 stimulation by violet light [6]. In EE, the phenotype of *Opn5$^{cre}$; Gnas$^{fl/fl}$* mice resembles that of the *Opn5$^{-/-}$* mouse: *Opn5* mutant mice show increases at 22˚C and 10˚C, while *Gnas* conditional mutant mice have an elevated MESOR at these temperatures. However, circadian analysis of *Opn5* mutant mice by Circacompare is required to assess the degree of similarity with *Opn5$^{cre}$; Gnas$^{fl/fl}$* mutant mice in EE rhythms. By contrast, *Opn5$^{cre}$; Gnas$^{fl/fl}$* mice show a temperature-dependent RER rhythm phenotype and a significant locomotor phenotype at 22˚C and above, while *Opn5*-null mice only have a slight RER phenotype by ANOVA at 10˚C and no observable difference in locomotion. Another difference between *Opn5$^{cre}$; Gnas$^{fl/fl}$* mice and *Opn5*-null mice is that *Opn5*-null mice consume more food and water than controls, while *Opn5$^{cre}$; Gnas$^{fl/fl}$* mice show little difference in either parameter. Circadian analysis should also be performed to determine rhythmic differences in these parameters. These phenotypic differences suggest that GNAS may function as a broad-based integrator of many GPCR inputs including OPN5, and OPN5 may use signaling mediators other than GNAS.

Additionally, *Opn5$^{cre}$; Gnas$^{fl/fl}$* mice and *Qrfp* mutant mice share a behavioral phenotype. QRFP is a neuropeptide expressed in the hypothalamus, and acts as a ligand for GPR103 [51]. Mice lacking *Qrfp* show several metabolic and behavioral phenotypes [51], including decreased wakefulness in the early part of the dark phase [51,52]. This specific deficit is consistent with the observed decrease in early-night locomotion in *Opn5$^{cre}$; Gnas$^{fl/fl}$* mice at 28˚C and 22˚C. This similarity suggests that QPLOT neurons regulate locomotor activity during the active phase, but the intersection between QRFP signaling and $G_{\alpha s}$ signaling is yet to be investigated.

Together, our metabolic studies suggest a role for $G_{\alpha s}$-signaling in QPLOT neurons in the daily pattern of metabolism. We found that deletion of *Gnas* in QPLOT neurons decreases nocturnal locomotion and perturbs metabolic rhythms at higher temperatures. Further studies will be required to identify the specific receptors responsible for these changes as well as the role of the SCN in regulating circadian metabolic patterns through the POA.

## Supporting information

**S1 Fig. Temperature challenge protocol.** Mice underwent a temperature protocol beginning at standard ambient temperature (22°C) for three days, followed by a two-day ramp to a cold challenge (10°C) for three days, and ending with a one-day ramp to thermoneutrality (28°C) for three days. Each temperature setting included one adaptation day and two days of recorded measurements (red outlines). 48-hour measurements were averaged into 24-hour profiles for statistical analysis. Each day consists of a 12h:12h light:dark cycle (respective yellow and gray shades).
(TIF)

**S2 Fig. Daily sleep time of *Gnas* conditional mutants and controls at different temperatures.** Sleep proportion in *Opn5^{cre};Gnas^{fl/fl}* (n = 9, blue) and *Opn5^{cre};Gnas^{+/+}* (n = 7, gray) mice was measured as a percent of each 15-minute interval in which the mouse was immobile after 40 seconds. Respective temperatures are indicated at the top of each column. For time of day on x-axes, t = 0 represents the start of the light phase at 6AM and t = 12 represents the start of the dark phase at 6PM (light:dark bar above each column). 24-hour sleep profiles (A-C) at thermoneutrality (A), standard ambient temperature (B), and during cold-stress (C). Data points in A-C are presented as mean± s.e.m. for 15-minute measurements and analyzed by two-way ANOVA for each 3-hour interval. Significant p-values are indicated above each interval. Cosinor curves (D-F) were generated by CircaCompare for thermoneutrality (D), standard ambient temperature (E), and during cold-stress (F). Estimates of circadian values and p-values (G-I) are listed under each cosinor curve.
(TIF)

**S1 File. Fig 1 Data.**
(XLSX)

**S2 File. Fig 2A–2C and 2G–2L Data.**
(XLSX)

**S3 File. Fig 2D–2F Data.**
(XLSX)

**S4 File. Fig 3 Data.**
(XLSX)

**S5 File. Fig 4 Data.**
(XLSX)

**S6 File. Fig 5 Data.**
(XLSX)

**S7 File. Fig 6 Data.**
(XLSX)

**S8 File. S2 Fig Data.**
(XLSX)

## Acknowledgments

We thank Paul Speeg for mouse colony management.

## Author Contributions

**Conceptualization:** Kevin D. Gaitonde, Richard A. Lang.

**Data curation:** Kevin D. Gaitonde.

**Formal analysis:** Kevin D. Gaitonde, Bala S. C. Koritala, Richard A. Lang.

**Funding acquisition:** David F. Smith, Richard A. Lang.

**Investigation:** Kevin D. Gaitonde, Mutahar Andrabi, Courtney A. Burger, Shane P. D'Souza, Shruti Vemaraju.

**Methodology:** Kevin D. Gaitonde, Shane P. D'Souza, Richard A. Lang.

**Project administration:** Richard A. Lang.

**Resources:** Shruti Vemaraju, Richard A. Lang.

**Software:** Shane P. D'Souza, Bala S. C. Koritala.

**Supervision:** Shruti Vemaraju, David F. Smith, Richard A. Lang.

**Visualization:** Kevin D. Gaitonde, Richard A. Lang.

**Writing – original draft:** Kevin D. Gaitonde, Richard A. Lang.

**Writing – review & editing:** Kevin D. Gaitonde, Mutahar Andrabi, Courtney A. Burger, Shane P. D'Souza, Shruti Vemaraju, Bala S. C. Koritala, David F. Smith, Richard A. Lang.

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
