## [Decision Letter · Decision Letter 0]

6 Feb 2023

PONE-D-22-33183Diurnal regulation of metabolism by GNAS in QPLOT neuronsPLOS ONE

Dear Dr. Lang,

Thank you for submitting your manuscript to PLOS ONE. After careful consideration, we feel that it has merit but does not fully meet PLOS ONE’s publication criteria as it currently stands. Therefore, we invite you to submit a revised version of the manuscript that addresses the points raised during the review process. While both reviewers acknowledge the potential interest in this piece of work, both have raised a number of concerns that need to be addressed in full before this can be reconsidered for publication. In particular I would advise the authors to address the serious concern that Opn5 is expressed in many other regions of the brain, where local effects of Gnas loss of function could account for the phenotypes that the authors currently ascribe exclusively to QPLOT neurons.

We look forward to receiving your revised manuscript.

Kind regards,

Nicholas Simon Foulkes, D.Phil

Academic Editor

PLOS ONE

Journal Requirements:

"The Lang lab participates in a sponsored research agreement between CCHMC and BIOS Lighting, Inc."

Reviewers' comments:

Reviewer's Responses to Questions

**Comments to the Author**

1. Is the manuscript technically sound, and do the data support the conclusions?

Reviewer #1: Partly

Reviewer #2: Partly

2. Has the statistical analysis been performed appropriately and rigorously? 

Reviewer #1: No

Reviewer #2: Yes

3. Have the authors made all data underlying the findings in their manuscript fully available?

Reviewer #1: Yes

Reviewer #2: Yes

4. Is the manuscript presented in an intelligible fashion and written in standard English?

Reviewer #1: Yes

Reviewer #2: Yes

5. Review Comments to the Author

Reviewer #1: Gaitonde et al. study the role of Gs signalling in the regulation of energy metabolism in mice. They show that mice carrying a deletion of Gnas (encoding the alpha subunit of Gs) in cells expressing neuropsin (Opn5) show reduced locomotor activity and subtle changes in the daily rhythms of energy metabolism as measured in an indirect calorimetry setup. They conclude that Opn5-positive neurons of the MPOA (Qplot neurons) regulate circadian (24-hour) rhythms of metabolism and activity.

This is an interesting paper, albeit reporting mostly negative findings, but several points should be addressed to substantiate the conclusions:

1. Gnas KO: No data validating specificity and efficiency of the Opn5-Cre-driven deletion of Gnas are provided. Is there recombination seen in ither brain areas expressing Opn5 (such as cortex, cerebellum, retina etc.) and could these affect metabolic phenotype? How efficient is Gnas recombination in Qplot neurons? Could the mild phenotype be the result of inefficient knock out?

2. A rescue experiment would be needed to confirm Qplot Gnas as the driver of the observed activity (and, potentially, other) phenotype, but this would go beyond the scope of this paper. This should be discussed, though.

3. CircaCompare uses cosine fits to describe rhythm parameters, but not all the rhythms analysed here show sinusoidal patterns. This may skew the results. Cosine fitting is probably ok for EE and RER (albeit under some conditions bimodal patterns are observed), but they clearly do not work for intake rhythms. Algorithms allowing for complex profiles would be more suitable such as RAIN or CircaN – but then they often do not allow for direct comparison of two conditions.

Minor points:

1. I would suggest avoiding acronyms in the title.

2. Would it be possible to extract sleep (or immobility as a proxy) data from the IR profiles? It remains unclear how Qplot Gs would affect activity, but a link to sleep has been made. Do mutant mice show more sleep during the dark phase compared to control animals?

Reviewer #2: In their manuscript “Diurnal regulation of metabolism by GNAS in QPLOT neurons”, the authors examine patterns of diurnal metabolism and behavior at three different ambient temperatures in mice carrying a CRE mediated deletion of a stimulatory G protein alpha subunit (Gnas), with CRE driven by regulatory sequences of Opn5 (neuropsin) from an Opn5 CRE knock-in locus. The authors examine energy expenditure (EE), respiratory exchange ratios (RER), food and water intake, as well as locomotion, analyzing the data both with two-way ANOVA (comparing 3 hour bins) and by CircaCompare (to determine rhythmicity, MESOR, amplitude and phase). ANOVA reveals significant differences only for locomotion at 28 and 22 °C, at the beginning (both temperatures) and end (only 28°C) of the dark phase. CircaCompare reveals more differences, also in the metabolic parameters. All changes, except perhaps for the altered locomotion patterns, appear to be minor in magnitude. The authors conclude that Gnas mediated signaling in QPLOT neurons of the preoptic area of the hypothalamus regulates daily patterns of metabolism in mice.

Data and data analysis of the manuscript are sound. An important issue, however, needs to be addressed regarding the experimental design and overall conclusion.

Specifically, the authors want to examine the effect of G-protein coupled receptor signaling in a specific subtype of neurons (QPLOT neurons) of the preoptic hypothalamus, which are characterized by a combined expression of specific markers, among them the opsin Opn5 (neuropsin). They use an allele driving CRE from an Opn5 knock-in locus to tissue-specifically excise a floxed Gnas allele. However, Opn5 is not only expressed in QPLOT neurons, but also in retinal ganglion cells (Nguyen et al., Nat Cell Biol 2019;21:420-429; D’Souza et al., J Comp Neurol. 2022;530:1247-1262) and apparently also in the cerebellum and cortex (Mouse ENCODE transcriptome data, https://www.ncbi.nlm.nih.gov/gene/353344). It is, therefore, possible that the observations made by the authors are not exclusively due to loss of function of Gnas in QPLOT neurons only, but might involve effects in other tissues as well that could directly or indirectly affect patterns of energy metabolism and behavior. This is especially important as the experiments were conducted under light-dark cycles, which might imply light dependent functions mediated by Opn5 expressing retinal ganglion cells.

Therefore, the authors either need to present additional data highlighting that indeed Gnas loss-of-function in their system is restricted to QPLOT neurons, or they should discuss the specificity issues in more detail and modify title and conclusion accordingly, referring rather to “Gnas function in Opn5 expressing tissues, including QPLOT neurons” or similarly.

Another issue is the rather subtle magnitude of the statistically significant effects observed for parameters other than locomotion, which are detected only by CircaCompare analysis. Could one main conclusion from the experiments be that metabolic patterns are relatively resilient to modification of Gnas mediated signaling in Opn5 expressing tissues, with other pathways playing redundant or more important roles? The authors may want to discuss such implications of their findings more explicitly.

Minor points:

Line 138: Why were only males used for indirect calometry experiments?

Line 271/272: “Interestingly, the of Opn5Cre;…”: add “the MESOR of ”

6. PLOS authors have the option to publish the peer review history of their article (what does this mean?). If published, this will include your full peer review and any attached files.

Reviewer #1: No

Reviewer #2: No

---

## [Author Response · Author response to Decision Letter 0]

24 Mar 2023

The following text can be found in the file "Response to Reviewers."

We thank the reviewers for their helpful comments in revising the manuscript PONE-D-22-33183. Except for re-assignment of figure numbers, line numbers are provided for all changes to the manuscript. The Revised Article with Changes Highlighted is referred to as the “Tracked Manuscript”; the Manuscript without highlighted changes is referred to as the “Manuscript.”

Reviewer #1: Gaitonde et al. study the role of Gs signalling in the regulation of energy metabolism in mice. They show that mice carrying a deletion of Gnas (encoding the alpha subunit of Gs) in cells expressing neuropsin (Opn5) show reduced locomotor activity and subtle changes in the daily rhythms of energy metabolism as measured in an indirect calorimetry setup. They conclude that Opn5-positive neurons of the MPOA (Qplot neurons) regulate circadian (24-hour) rhythms of metabolism and activity.

This is an interesting paper, albeit reporting mostly negative findings, but several points should be addressed to substantiate the conclusions:

1. Gnas KO: No data validating specificity and efficiency of the Opn5-Cre-driven deletion of Gnas are provided. Is there recombination seen in ither brain areas expressing Opn5 (such as cortex, cerebellum, retina etc.) and could these affect metabolic phenotype? How efficient is Gnas recombination in Qplot neurons? Could the mild phenotype be the result of inefficient knock out?

In previous research[1–3] reliable Opn5 expression in the adult central nervous system using Opn5Cre is only detected in the hypothalamic preoptic area and a subset of retinal ganglion cells. Hypothalamic Opn5 neurons show marked effects on metabolism even in the absence of violet light and OPN5 activity. Furthermore, enucleated mice, which eliminates any influence of retinal Opn5, show similar metabolic perturbations. Retinal Opn5 is confined to a small subset of all retinal ganglion cells, and very few of these Opn5-expressing RGCs project to the suprachiasmatic nucleus, which controls daily metabolic cycles. Therefore, it is very unlikely that retinal Opn5 cells contribute to circadian variation. These reasons are inserted into the second paragraph of the Discussion (Tracked manuscript: lines 532-533, 539-543; Manuscript: lines 468-469, 472-476).

We agree that the efficiency of Gnas deletion in Opn5 cells is important for the validation of our knockout model and for the context of our observed metabolic differences. We have inserted a new figure (Fig 1) in which we demonstrate that signal from the Gnas transcript is reduced in Opn5 cells of the Gnas conditional mutant. Subsequent figures were renumbered. The new text regarding the new Figure 1 is included in the Methods (Tracked manuscript: lines 149-186, 206-208; Manuscript: lines 148-184, 204-206), Results (Tracked manuscript: lines 217-257; Manuscript: 215-253), and Discussion (Tracked manuscript: lines 543-545; Manuscript: lines 477-479). We have also added Shane P. D’Souza as a co-author for involvement in this experiment on the Title Page (Tracked manuscript and Manuscript: lines 6-7) and Author Contributions (Tracked manuscript: lines 653, 654, 657, 661; Manuscript: lines 587, 588, 591, 595).

2. A rescue experiment would be needed to confirm Qplot Gnas as the driver of the observed activity (and, potentially, other) phenotype, but this would go beyond the scope of this paper. This should be discussed, though.

We agree that this experiment would be valuable in determining the role of Gs signaling in QPLOT neurons in circadian variation. We have added this experiment into the Discussion (Tracked manuscript: lines 552-555; Manuscript: 486-489) to finish our explanation of how the deletion of Gnas impairs cAMP signaling.

3. CircaCompare uses cosine fits to describe rhythm parameters, but not all the rhythms analysed here show sinusoidal patterns. This may skew the results. Cosine fitting is probably ok for EE and RER (albeit under some conditions bimodal patterns are observed), but they clearly do not work for intake rhythms. Algorithms allowing for complex profiles would be more suitable such as RAIN or CircaN – but then they often do not allow for direct comparison of two conditions.

We totally agree with the reviewer’s recommendation that RAIN or CircaN may be more appropriate. As the reviewer mentioned, these algorithms are limited in their ability to evaluate the presence or absence of rhythmicity and cannot be used to compare differences of rhythms between conditions. As our research focuses on comparing two conditions (Opn5cre;Gnas+/+ and Opn5cre;Gnasfl/fl mice), we employed the available widely accepted CircaCompare [4] to compare the rhythmicity and related properties such as differences in amplitude, acrophase, and MESOR. The authors of Circacompare have stated in the original paper that the requirement for CircaCompare to be used for a data set is rhythmicity, and we included the test for rhythmicity in our analysis (Fig 2J-L, Figs 3-6 G-I, Supp Fig 2 G-I, first two lines each), demonstrating that each parameter is significantly rhythmic.

Minor points:

1. I would suggest avoiding acronyms in the title.

While it would be difficult to expand QPLOT, given that this is a summary of important biomarkers that define these neurons, we have changed “GNAS” to “Gs-alpha” on the Title Page (Tracked Manuscript and Manuscript: line 1). We also inserted “hypothalamic” in front of “QPLOT” so the location of these neurons is immediately known (Title Page line 1).

2. Would it be possible to extract sleep (or immobility as a proxy) data from the IR profiles? It remains unclear how Qplot Gs would affect activity, but a link to sleep has been made. Do mutant mice show more sleep during the dark phase compared to control animals?

We have extracted sleep data from the same mice tested in all of the other experiments and included another supplemental figure for sleep analysis (Supp Fig 2). Sleep was measured as a proportion of each time interval in which the mouse was inactive after 40 seconds of inactivity. New text for sleep has been added to the Methods (Tracked Manuscript: lines 201-202; Manuscript: lines 199-200), Results (Tracked Manuscript: lines 519-524; Manuscript: lines 455-460), Discussion (Tracked Manuscript: lines 567 and 572; Manuscript: lines 501 and 572), and Supplemental Figure Legends (Tracked Manuscript: lines 827-839; Manuscript: lines 761-773). 

Reviewer #2: In their manuscript “Diurnal regulation of metabolism by GNAS in QPLOT neurons”, the authors examine patterns of diurnal metabolism and behavior at three different ambient temperatures in mice carrying a CRE mediated deletion of a stimulatory G protein alpha subunit (Gnas), with CRE driven by regulatory sequences of Opn5 (neuropsin) from an Opn5 CRE knock-in locus. The authors examine energy expenditure (EE), respiratory exchange ratios (RER), food and water intake, as well as locomotion, analyzing the data both with two-way ANOVA (comparing 3 hour bins) and by CircaCompare (to determine rhythmicity, MESOR, amplitude and phase). ANOVA reveals significant differences only for locomotion at 28 and 22 °C, at the beginning (both temperatures) and end (only 28°C) of the dark phase. CircaCompare reveals more differences, also in the metabolic parameters. All changes, except perhaps for the altered locomotion patterns, appear to be minor in magnitude. The authors conclude that Gnas mediated signaling in QPLOT neurons of the preoptic area of the hypothalamus regulates daily patterns of metabolism in mice.

Data and data analysis of the manuscript are sound. An important issue, however, needs to be addressed regarding the experimental design and overall conclusion.

Specifically, the authors want to examine the effect of G-protein coupled receptor signaling in a specific subtype of neurons (QPLOT neurons) of the preoptic hypothalamus, which are characterized by a combined expression of specific markers, among them the opsin Opn5 (neuropsin). They use an allele driving CRE from an Opn5 knock-in locus to tissue-specifically excise a floxed Gnas allele. However, Opn5 is not only expressed in QPLOT neurons, but also in retinal ganglion cells (Nguyen et al., Nat Cell Biol 2019;21:420-429; D’Souza et al., J Comp Neurol. 2022;530:1247-1262) and apparently also in the cerebellum and cortex (Mouse ENCODE transcriptome data, https://www.ncbi.nlm.nih.gov/gene/353344). It is, therefore, possible that the observations made by the authors are not exclusively due to loss of function of Gnas in QPLOT neurons only, but might involve effects in other tissues as well that could directly or indirectly affect patterns of energy metabolism and behavior. This is especially important as the experiments were conducted under light-dark cycles, which might imply light dependent functions mediated by Opn5 expressing retinal ganglion cells.

Therefore, the authors either need to present additional data highlighting that indeed Gnas loss-of-function in their system is restricted to QPLOT neurons, or they should discuss the specificity issues in more detail and modify title and conclusion accordingly, referring rather to “Gnas function in Opn5 expressing tissues, including QPLOT neurons” or similarly.

In the adult mouse central nervous system, Opn5 expression through Opn5cre is reliably detected in the hypothalamus and some retinal ganglion cells (RGCs) [1–3]. Opn5cre is expressed in a small subset of RGCs, and these Opn5 RGCs do not project to the SCN. Furthermore, enucleated mice, which no longer receive retinal input, demonstrate similar metabolic profiles as global Opn5 mutants [1]. Opn5cre is not detected in the cerebellum or cortex [1]. We have added this explanation to the Discussion (Tracked manuscript: lines 532-533, 539-543; Manuscript: lines 468-469, 472-476).

Another issue is the rather subtle magnitude of the statistically significant effects observed for parameters other than locomotion, which are detected only by CircaCompare analysis. Could one main conclusion from the experiments be that metabolic patterns are relatively resilient to modification of Gnas mediated signaling in Opn5 expressing tissues, with other pathways playing redundant or more important roles? The authors may want to discuss such implications of their findings more explicitly.

We agree that Gs signaling is only one of several signaling pathways in QPLOT neurons that regulate metabolism. Because QPLOT neurons can exert profound effects on metabolism and express several important receptors that are GPCRs (EP3, Opn5, NK3R) or non-GPCRs (LEPR), these neurons must be well-regulated so that a stimulus can produce an appropriate adaptation without wasting energy. The involvement of multiple pathways may suggest why the deletion of Gs signaling is relatively modest. This is added to the Discussion (Tracked Manuscript: lines 555-562; Manuscript: lines 489-496).

Minor points:

Line 138: Why were only males used for indirect calometry experiments?

While we considered using males and females for these experiments, young adult female mice have regular estrous cycles which can produce day-to-day changes in metabolic parameters[5]. This would increase the standard error in our daily profiles.

Line 271/272: “Interestingly, the of Opn5Cre;…”: add “the MESOR of ”

 We have corrected this mistake (Tracked Manuscript: line 380; Manuscript: line 341). Thank you for noticing it.

Other changes:

1. We removed “expenditure” from the new Fig 6 caption (Tracked Manuscript: line 495; Manuscript: line 436).

2. We inserted (F) in the figure legends for Figs 3-6. “Cosinor curves … and during cold-stress (F)” (Tracked Manuscript: lines 408, 445, 474, 500; Manuscript: lines 361, 393, 417, 441). 

1. Zhang KX, D’Souza S, Upton BA, Kernodle S, Vemaraju S, Nayak G, et al. Violet-light suppression of thermogenesis by opsin 5 hypothalamic neurons. Nature. 2020;585(7825). 

2. D’Souza SP, Swygart DI, Wienbar SR, Upton BA, Zhang KX, Mackin RD, et al. Retinal patterns and the cellular repertoire of neuropsin (Opn5) retinal ganglion cells. J Comp Neurol [Internet]. 2021 Nov 7; Available from: https://onlinelibrary.wiley.com/doi/10.1002/cne.25272

3. Nguyen MTT, Vemaraju S, Nayak G, Odaka Y, Buhr ED, Alonzo N, et al. An opsin 5–dopamine pathway mediates light-dependent vascular development in the eye. Nat Cell Biol. 2019 Apr 1;21(4):420–9. 

4. Parsons R, Parsons R, Garner N, Oster H, Rawashdeh O. CircaCompare: A method to estimate and statistically support differences in mesor, amplitude and phase, between circadian rhythms. Bioinformatics. 2020 Feb 15;36(4):1208–12. 

5. Mauvais-Jarvis F, Clegg DJ, Hevener AL. The role of estrogens in control of energy balance and glucose homeostasis. Vol. 34, Endocrine Reviews. 2013. p. 309–38.

---

## [Decision Letter · Decision Letter 1]

10 Apr 2023

Diurnal regulation of metabolism by Gs-alpha in hypothalamic QPLOT neurons

PONE-D-22-33183R1

Dear Dr. Lang,

We’re pleased to inform you that your manuscript has been judged scientifically suitable for publication and will be formally accepted for publication once it meets all outstanding technical requirements.

Kind regards,

Nicholas Simon Foulkes, D.Phil

Academic Editor

PLOS ONE

Additional Editor Comments (optional):

Reviewers' comments:

Reviewer's Responses to Questions

**Comments to the Author**

1. If the authors have adequately addressed your comments raised in a previous round of review and you feel that this manuscript is now acceptable for publication, you may indicate that here to bypass the “Comments to the Author” section, enter your conflict of interest statement in the “Confidential to Editor” section, and submit your "Accept" recommendation.

Reviewer #1: All comments have been addressed

Reviewer #2: All comments have been addressed

2. Is the manuscript technically sound, and do the data support the conclusions?

Reviewer #1: Yes

Reviewer #2: (No Response)

3. Has the statistical analysis been performed appropriately and rigorously? 

Reviewer #1: Yes

Reviewer #2: (No Response)

4. Have the authors made all data underlying the findings in their manuscript fully available?

Reviewer #1: Yes

Reviewer #2: (No Response)

5. Is the manuscript presented in an intelligible fashion and written in standard English?

Reviewer #1: Yes

Reviewer #2: (No Response)

6. Review Comments to the Author

Reviewer #1: Congrats on this nice story. I have no further comments or questions. Not sure what else I should wirite here given that all my concerns have been addressed.

Reviewer #2: (No Response)

7. PLOS authors have the option to publish the peer review history of their article (what does this mean?). If published, this will include your full peer review and any attached files.

Reviewer #1: No

Reviewer #2: No

---

## [Editor Report · Acceptance letter]

26 Apr 2023

PONE-D-22-33183R1 

Diurnal regulation of metabolism by G_s_-alpha in hypothalamic QPLOT neurons 

Dear Dr. Lang:

I'm pleased to inform you that your manuscript has been deemed suitable for publication in PLOS ONE. Congratulations! Your manuscript is now with our production department. 

Kind regards, 

on behalf of

Dr. Nicholas Simon Foulkes 

Academic Editor

PLOS ONE